# Radiomic and Volumetric Measurements as Clinical Trial Endpoints—A Comprehensive Review

**DOI:** 10.3390/cancers14205076

**Published:** 2022-10-17

**Authors:** Ionut-Gabriel Funingana, Pubudu Piyatissa, Marika Reinius, Cathal McCague, Bristi Basu, Evis Sala

**Affiliations:** 1Cambridge University Hospitals NHS Foundation Trust, Cambridge CB2 0QQ, UK; 2Department of Oncology, University of Cambridge, Cambridge CB2 0XZ, UK; 3Cancer Research UK Cambridge Centre, University of Cambridge, Cambridge CB2 0RE, UK; 4Cancer Research UK Cambridge Institute, University of Cambridge, Cambridge CB2 0RE, UK; 5School of Clinical Medicine, University of Cambridge, Cambridge CB2 0SP, UK; 6Department of Radiology, University of Cambridge, Cambridge CB2 0QQ, UK

**Keywords:** imaging biomarkers, clinical trials, surrogate endpoints, volumetric, radiomics, data integration

## Abstract

**Simple Summary:**

The extraction of quantitative data from standard-of-care imaging modalities offers opportunities to improve the relevance and salience of imaging biomarkers used in drug development. This review aims to identify the challenges and opportunities for discovering new imaging-based biomarkers based on radiomic and volumetric assessment in the single-site solid tumor sites: breast cancer, rectal cancer, lung cancer and glioblastoma. Developing approaches to harmonize three essential areas: segmentation, validation and data sharing may expedite regulatory approval and adoption of novel cancer imaging biomarkers.

**Abstract:**

Clinical trials for oncology drug development have long relied on surrogate outcome biomarkers that assess changes in tumor burden to accelerate drug registration (i.e., Response Evaluation Criteria in Solid Tumors version 1.1 (RECIST v1.1) criteria). Drug-induced reduction in tumor size represents an imperfect surrogate marker for drug activity and yet a radiologically determined objective response rate is a widely used endpoint for Phase 2 trials. With the addition of therapies targeting complex biological systems such as immune system and DNA damage repair pathways, incorporation of integrative response and outcome biomarkers may add more predictive value. We performed a review of the relevant literature in four representative tumor types (breast cancer, rectal cancer, lung cancer and glioblastoma) to assess the preparedness of volumetric and radiomics metrics as clinical trial endpoints. We identified three key areas—segmentation, validation and data sharing strategies—where concerted efforts are required to enable progress of volumetric- and radiomics-based clinical trial endpoints for wider clinical implementation.

## 1. Introduction

Clinical trials rely on pre-defined endpoints to evaluate the efficacy of a given medical product. Instead of directly measuring the clinical outcome, the process can be accelerated by the use of alternative markers that identify therapeutic response early during treatment [1]. These ‘surrogate endpoints’, as defined by the FDA section 507(e)(9), comprise laboratory measurements, radiographic images, physical signs or other measures, that are not themselves a direct measurement of clinical benefit, but which are at least reasonably likely to predict clinical benefit from a drug or biological product [2]. 

All FDA approved imaging-based surrogate endpoints for treatment response assessment in solid tumors—namely objective response rate (ORR), progression-free survival (PFS), metastasis-free survival, disease-free survival (DFS) and event-free survival (EFS))—currently employ the so-called Response Evaluation Criteria in Solid Tumors version 1.1 (RECIST v1.1) response criteria [2]. RECIST v1.1 evaluates the change in tumor burden by means of a series of standardized rules based on the unidimensional size of the lesions [3]. These criteria are mechanism-agnostic, and relationships between changes in tumor size and survival benefit vary according to tumor type [2]. 

Other widely employed surrogate endpoints include histopathological changes following drug exposure, such as pathological complete response (pCR) for patients with breast cancer [2]. Integration of more complex multimodal data, including data extraction using quantitative features from standard of care imaging, may improve the clinical relevance of surrogate endpoints during evaluation of novel therapies targeting complex biological systems (such as the immune system). There are broadly two different approaches to extracting quantitative features from imaging data: hand-crafted feature generation, for example, using one of many open source radiomic tools (MIRP, S-IBEX, RaCaT, SERA, PyRadiomics and RadiomiCRO), and deep learning approaches where features are generated as part of the learning process [4,5]. 

Prior to the implementation of RECIST v1.1, volumetric-based imaging biomarkers were considered as an alternative to assessment by unidimensional measurement, but the lack of standardization and evidence to support the transition led the RECIST working group to abandon the idea [3]. However, volumetric response measurement offers some advantages compared to RECIST v1.1, including reduction of inter-reader variability [6] and increased likelihood of correctly identifying pseudo-progression [7]. Several prospective studies suggest that volumetric measurements performed better than planar RECIST v1.1 assessment [8,9]. In contrast to volumetric assessment, it has been suggested that radiomic features may add more biological value to the standard dimensional measurements [4]. 

A Cancer Research UK—European Organisation for Research and Treatment of Cancer (CRUK-EORTC) consensus statement has summarized the roadmap for clinical translation of imaging biomarkers and highlighted the need for the candidate biomarkers to close two ‘translational gaps’: the validation as robust medical research tools; and integration into routine patient care [10]. Within this roadmap, imaging biomarkers may play a role in the discovery phase (e.g., non-clinical studies) and in early technical, biological and clinical validation. Significant efforts have been made internationally to harmonize all stages within a radiomics pipeline from data extraction and curation to operating procedures for standardization and reproducibility, interpretability, generalizability and regulatory approval. These have resulted in initiatives such as the Radiomic Quality Score (RQS), Transparent Reporting of a multivariable prediction model for Individual Prognosis or Diagnosis (TRIPOD) and Image Biomarker Standardization Initiative (IBSI)) [11,12,13]. The steps required to address the second translational gap to bridge the space to routine patient care, involve incorporation into prospective multi-site clinical trials with further qualification and technical validation [10]. 

In order to assess the status of volumetric and radiomics-based biomarkers as potential clinical trial endpoints in oncological studies, this review aims to synthesize the literature of computational modelling of imaging features in predicting treatment response and outcome. To capture a representative summary of the current status of this expanding field, we focus on four key tumor types with a significant body of published data, in order to provide a detailed overview of methodological considerations and reporting of model performance. This informs our discussion of key areas that require improvement in order to close the ‘translational gaps’ and prepare volumetric and radiomics-based biomarkers for prime time.

## 2. Article Search Strategy and Study Selection

A literature search was performed on 1 July 2021 to identify primary research papers that have used radiomics or volumetric approaches to study predictive or prognostic models. Articles published between 1 January 2015 and 30 June 2021 were included (Figure 1). The main statements of the inclusion and exclusion criteria for the studies entered into the review are related to considered population (patients with solid tumor diagnosis), type of interventions (volumetric and/or radiomic analyses) and the comparators and outcomes measured against (RECIST response or pathologic response). To identify published computational/machine learning models in the oncological imaging literature, we used the following regular expressions to query the PubMed database: cancer/tumor/tumor/oncology/oncological/malignancy, radiomic/radiogenomic/volumetric/computational/machine/learning/deep learning/framework, predict/prediction, response/pathological and survival/outcome/PFS (see Appendix A for PubMed Query box expression). 

### Data Extraction and Relevance Rating

The abstracts of the identified papers were screened using a list of inclusion and exclusion criteria (see Appendix A) using the open-source web-based platform https://www.rayyan.ai/ (accessed on 1 July 2021) [14]. Abstract screening was initially performed by one author (IGF). The papers were included in the three groups of relevance: included, excluded and maybe (the last group mainly for papers with abstracts of uncertain relevance due to unclear study design). 

Four reviewers assessed the articles identified by the abstract screening process according to eight domains (see Appendix A for full details). The domains covered were patient characteristics; study design and dataset; imaging details; non-radiological data types; algorithm details; model performance; generalizability/reproducibility; and regulatory approval (e.g., FDA approval) (see Appendix A). 

## 3. Results

### 3.1. Overview of Included Studies 

The relevance rating of the published models was heterogeneous with only 27.3% of the total number of papers passing the screening stage, and only 124/168 papers from the first stage were eligible for the final review part (Figure 1). 

The total number of published papers increased each year (Appendix A). The highest proportion of eligible papers were identified for the single-site solid tumor sites: breast cancer, rectal cancer, lung cancer and glioblastoma (Figure 2) and therefore, the review focused on use of radiomics and volumetric endpoints for these four tumor cohorts. 

#### 3.1.1. Breast Cancer (BC)

After the initial abstract screening, 43 breast cancer papers were included in the final analysis. The evaluation of the full manuscripts led to the exclusion of nine publications for the following reasons: radiomic analysis of distant metastatic sites (*n* = 2), small sample size (*n* = 2), non-predictive or prognostic endpoints (*n* = 3), in silico model development (*n* = 1) and non-imaging modality (*n* = 1). The remaining articles (*n* = 34) were included in the final analysis [15,16,17,18,19,20,21,22,23,24,25,26,27,28,29,30,31,32,33,34,35,36,37,38,39,40,41,42,43,44,45,46,47,48]. 

#### 3.1.2. Rectal Cancer (RC)

After the initial abstract screening, 34 papers were included for further analysis; 19 were excluded for the following reasons: relating to non-rectal cancers (*n* = 1), insufficient information provided on how region of interest (ROI) segmentation was performed (*n* = 2), mixed or unspecified histopathology (*n* = 10), insufficient detail on the acquisition parameters (*n* = 4), no radiomic or volumetric analysis performed (*n* = 1) and non-predictive association study (*n* = 1). The remaining articles (*n* = 15) related to locally advanced rectal cancer which were all histologically confirmed as adenocarcinoma [49,50,51,52,53,54,55,56,57,58,59,60,61,62,63]. 

#### 3.1.3. Lung Cancer (LC)

The initial abstract screen identified 49 lung cancer papers as relevant for manual review. Of these, five were excluded due to: non-imaging modality (*n* = 1), non-predictive or prognostic endpoints (*n* = 2), radiomic analysis of tumors metastatic to the lung (*n* = 1), treatment strategy not recorded (*n* = 1). This yielded a total of 44 papers [64,65,66,67,68,69,70,71,72,73,74,75,76,77,78,79,80,81,82,83,84,85,86,87,88,89,90,91,92,93,94,95,96,97,98,99,100,101,102,103,104,105,106,107].

#### 3.1.4. Glioblastoma Multiforme (GBM)

A total of 42 articles on glioblastoma multiforme were identified by the abstract screening strategy described; 11 were excluded due to: investigation of cerebral blood volume rather than tumor volume (*n* = 3); neither radiomic nor volumetric analyses performed (*n* = 3); limited imaging or other methodological details (*n* = 2); mixed or unspecified histopathology (*n* = 2); detection of progressive disease on serial imaging rather than prediction of future response (*n* = 1). The remaining articles (*n* = 31) were included in the final analysis [108,109,110,111,112,113,114,115,116,117,118,119,120,121,122,123,124,125,126,127,128,129,130,131,132,133,134,135,136,137,138]. 

### 3.2. Study Design 

The subgroup analyses revealed that the majority of the studies analyzed retrospective datasets: BC (76%), RC (80%), LC (89%) and GBM (80%), with the remainder included prospective collected data of clinical trials. A limited proportion of imaging data was collected as part of a multi-institution collaboration for the included tumor types: BC (23.5%), RC (40%), LC (25.0%) and GBM (22.5%) (See Table 1).

#### 3.2.1. Breast Cancer

The majority of the papers (*n* = 26) were retrospective studies, and imaging data collected during the conduct of two prospective clinical trials was examined. Five papers included the analysis of imaging data from the ACRIN 6657/ISPY-1 (NCT00033397) clinical trial [18,24,25,32,36] and one paper from the ASAINT (NCT02599974) study [30]. Over half (5/8 studies) of the prospective studies comprised multi-institution collaborations and 31 out of 34 analyses utilized imaging data of neoadjuvantly-treated patients, two studies utilized imaging data of primary surgery cohorts and one study was of a second line treatment population.

#### 3.2.2. Rectal Cancer

All studies related to first line treatment. The majority were retrospective (*n* = 12), and the remainder prospective in design, including, NCT01171300 [53] and FOWARC NCT01211210 [54]. Six of the papers were multi-institution studies, with the remainder involving a single institution.

#### 3.2.3. Lung Cancer

Of the 44 papers reviewed, only six papers that were associated with one or more clinical trials: (NCT00533949, NCT02136355, NCT00087438, NCT00181545, NCT00181506, NCT00572325, NCT00573040, NCT01166204, NCT01084785 and NCT01936571) [78,79,82,95,103]. The majority of the studies (39/44) were retrospective and one third (17/44) of the studies for related to first line treatment. 

#### 3.2.4. Glioblastoma Multiforme

Study designs were mostly retrospective (*n* = 25); however, three studies with modest sample sizes (range 15–54) were conducted prospectively (including GLIAA-Pilot/DRKS00000633 and NCT02329795) [135,138]. A further two studies involved retrospective analyses of larger prospectively collected clinical trial datasets (NCT01089868 and DIRECTOR/NCT 00941460) [112,113], as well as one single-institution retrospective study including a prospective test-retest cohort [131]. Most analyses pertained to the first-line setting (*n* = 26), and fewer studies investigated recurrence images (*n* = 5).

### 3.3. Type of Imaging Modality and Strategy of Segmentation

The volumetric and/or radiomic features were extracted using a semi-automated or automated pipeline for 10/34 of BC (29.4%), 2/15 of RC (13.3%), 17/44 of LC papers (38.6%) and 18/31 of GBM (58.0%). Most of the studies assessed the performance of handcrafted features applying classical statistical methods or machine learning models, with only eight studies employing deep learning approaches. 

#### 3.3.1. Breast Cancer

The majority of papers (25/34) evaluated MRI imaging-based data. Four studies were of FDG-PET/CT scans, three studies were of ultrasound scans and two were of CT scans. Automated or semi-automated data extraction strategies were applied in 10 studies, but in the other 24 studies, a manual data curation step was included upstream to imaging features extraction. On average, two assessors per study were used to segment/assess the raw imaging data manually (median = 2, mean = 1.9, range 1–9), with more assessors required for the multi-institution prospective studies e.g., for the ACRIN 6657/ISPY-1 (NCT00033397) study [18,24,25,32,36], nine assessors in total (one per investigation site) were used.

#### 3.3.2. Rectal Cancer

MRI is the preferred imaging modality for the local staging of rectal cancer [139], and this was reflected in the included papers, with the majority (*n* = 10) of studies reporting on MRI alone, or in combination with other imaging modalities (MRI and CT, *n* = 1; MRI and PET-CT *n* = 2; PET-MRI and CT *n* = 1). A single study used CT as its sole imaging modality [57]. All of the included studies clearly outlined the parameters for how the imaging was acquired, and some (*n* = 4) included detail on the pre-processing steps performed prior to image analysis. The method of the ROI segmentation in the 15 included studies varied: 13 included manual segmentations alone, 1 study contained a mix of manual and semi-automated segmentations, and 1 study contained only semi-automated segmentations. The number of assessors per study varied from 1 to 4, and were either radiologists, radio-oncologists or researchers experienced in segmentation.

#### 3.3.3. Lung Cancer

The majority of papers (29/44) evaluated CT images, with 15/44 evaluating PET CT images and 1/44 evaluating Cone beam CT images. The image acquisition method, including scanner model, slice thickness and use of contrast, was reported fully in 26 papers. Segmentation was reported as automated, semi-automated or manual by 4, 18 and 22 studies respectively. Of the studies using semi-automated or manual segmentation, the number of assessors ranged between 1 and 3 (median = 1), with 20 papers not reporting the number of assessors. Tumor margin and peritumor voxels are salient features for predicting treatment failure [79], but none of the papers assessed carried out robustness studies to assess inter-reader variability.

#### 3.3.4. Glioblastoma Multiforme

MRI is the standard radiological modality in GBM management [140], and was the sole imaging type in 29 of 31 studies. Other modalities investigated were PET-MRI (*n* = 1) and FET-PET (*n* = 1). In terms of image analysis pipelines, the segmentation approach was reported as automated, semi-automated or manual by 7, 11 and 9 studies, respectively, and was not specified in 4 manuscripts. Of the 20 studies using semi-automated or manual segmentation, the assessors ranged between 1 and 3 (median = 2) radiologists, oncologists or other expert researchers. A total of 26 studies used pre-treatment scans alone, whilst five used pre and post-treatment images.

### 3.4. Type of Algorithms and Primary Endpoints 

The most commonly used methods were Cox regression (*n* = 40), logistic regression (*n* = 20) and random forest (*n* = 15) and support vector machine (SVM) (*n* = 16). Other machine learning models were implemented for the predictive or prognostic analysis of imaging features in 10 papers (See Table 2).

#### 3.4.1. Breast Cancer

The algorithms implemented for radiomic or volumetric analyses were heterogeneous with 12 different types of algorithms deployed: 3D mathematical model (*n* = 1), extended Tofts–Kety model (ETK) (*n* = 1), convolutional neural network (CNN) algorithm (*n* = 3), Cox regression model (*n* = 6), Histogram (*n* = 1), Jacobian map (*n* = 1), linear discriminant analysis (*n* = 1), logistic regression (*n* = 6), other machine learning models (*n* = 1), nomogram (*n* = 5), random forest (*n* = 2) and SVM (*n* = 6). Additional computational studies were carried out for four out of six SVM analyses, two out of six logistic regression analyses and one out of five nomogram analyses. The additional computational models encompass Fisher’s linear discriminant (FLD), k-nearest neighbour (KNN), stochastic gradient descent (SGD), decision tree adaptive boosting (AdaBoost) and extreme gradient boosting (XGBoost) methods.

The pCR scores were the primary endpoint for most studies, either alone (*n* = 22) or in combination with survival endpoints (*n* = 5) or ORR by RECIST v1.1 (*n* = 1). The 6 remaining computational models tested either the prognostic value of the analyses against survival endpoints (disease-free survival for *n* = 4 or overall survival for *n* = 1) or the predictive value against a non-standardized imaging-based endpoint measuring the Kendall correlation coefficients (KCC) between model-predicted tumor response and the observed values at the time of the final scan (*n* = 1). 

#### 3.4.2. Rectal Cancer

A variety of approaches were applied to the analysis of the collected data. Overall, 11 studies used a single model (logistic regression: *n* = 6; linear regression: *n* = 2; partial least square (PLS) regression *n* = 1; LASSO Cox regression model *n* = 1; and SVM *n* = 1). Two studies applied multiple models to the analysis; the first (using logistic regression, SVM, and gradient boosting machine (GBM)) tested its models in parallel [54], the second used its first model (an SVM) to create a radiomic signature which was then ensembled with a second multivariate logistic regression model to create a radiomic model. The performances of both the radiomic signature and radiomic model were tested with an independent validation cohort [62]. Two studies used a deep learning-based approach. 

The majority (*n* = 14) of the studies used pathological complete response, or disease response by tumor regression grade (TRG), as an endpoint and a single study used disease free survival (DFS) [58].

#### 3.4.3. Lung Cancer

The algorithms implemented for the analysis of collected data were varied. Most applied a single model: 21 studies utilized logistic (*n* = 1)/linear (*n* = 1)/COX regression (*n* = 18}/Recursive partitioning (*n* = 1) approaches. Seven studies utilized machine learning approaches including support vector machine (*n* = 2), random forest (*n* = 3) and neural network (*n* = 2). Of the papers assessing multiple models (*n* = 16), approaches included logistic, linear and COX regression, unsupervised clustering, random forest, neural network, support vector machine and nomograms.

The primary endpoints assessed were either OS alone (*n* = 16), in combination with PFS (*n* = 6), or DFS (*n* = 3). The remaining papers assessed DFS (*n* = 5), PFS (*n* = 10), cause specific survival (*n* = 1), or other measurements of tumor response (*n* = 3) [90,103,104], including one paper measuring a pCR. While some papers utilized RECIST to assess tumor response to treatment (*n* = 8), Yang et al. [103] were the only group to compare the predictive ability of their imaging biomarker against that of RECIST or PERCIST. They found that their recursive partitioning analysis model achieved better pCR prediction (Concordance: 0.92; *p* = 0.03) than RECIST (Concordance: 0.54) or PERCIST (Concordance 0.58).

#### 3.4.4. Glioblastoma Multiforme

Most (*n* = 29) studies applied a single model: 17 performed logistic/linear/Cox regression, one used apparent diffusion coefficient histogram analysis, and 11 used machine learning approaches (random forest: *n* = 6, support vector machine: *n* = 5). Of two studies that used multiple machine learning algorithms, one reported that XGBoost achieved superior performance to a random forest model [110]. The other demonstrated that linear regression-based genomic and radiogenomic prediction models outperformed counterparts that used RF, SVM, artificial neural network and gradient boosting methods [134].

Predicted endpoints were predominantly OS alone (*n* = 18), or both PFS and OS (*n* = 8). Other studies predicted PFS alone (*n* = 1); PFS and site of recurrence (*n* = 1); time to progression, OS and site of recurrence (*n* = 1) [135]; and site of recurrence (*n* = 2) [124,138].

### 3.5. Integration of Radiomics with Genomics and Multi-Omics

Imaging features were integrated with validated and prognostic and/or predictive markers (e.g., clinical, histological or molecular data) in 20/34 of BC models (58.8%), 9/15 of RC models (60.0%), 15/44 of LC models (34.0%) and 19/31 of GBM models (61.2%) in the articles included in this review. However, exploratory genomics or transcriptomics (including longitudinal ctDNA and RNA sequencing) or other molecular datapoints were integrated in a minority of the analyses (BC 1/34, RC 1/15, LC 5/44 and GBM 2/31). 

#### 3.5.1. Breast Cancer

Molecular properties (ER status by IHC and HER2 status by ISH/IHC) were reported for 26/34 (76%) of the manuscripts. Three papers included pre-specified molecular inclusion criteria, two included only HER2+ breast cancer population and one was for triple-negative breast cancer (TNBC). Clinical and molecular factors with established prognostic value were integrated with radiomics or volumetric features for the majority (20/34 studies—58.8%) of the models. Two radiomics studies integrated other molecular data points (RNA sequencing of TCIA cohort or lncRNA sequencing data) [27,32].

#### 3.5.2. Rectal Cancer

A total of 10 studies used models which included clinical data, and only one of the 15 studies included molecular data, in the form of circulating tumor cells (CTCs) and extracellular vesicles (EVs) such as cancer microparticles (MPs) [63].

#### 3.5.3. Lung Cancer

In all, 95% (*n* = 42) of papers recorded clinical details such as Eastern Cooperative Oncology Group (ECOG) performance status, and histology, and (*n* = 12) papers incorporated this data into multiomics models. Additionally, 9 papers also used deltaomic biomarkers, quantifying a change in imaging signature over time, and 2 of the 12 papers combined both multiomics and deltaomic data [91,107]. The molecular properties of lung tumors (*EGFR* pathogenic variants, *ROS* or *ALK* genes rearrangements, PD-L1 expression) were studied (*n* = 8), and used as inclusion criteria (*n* = 5), or assessed for predictive value (*n* = 4). 

#### 3.5.4. Glioblastoma Multiforme

Fewer than half (*n* = 12) of the publications reviewed included imaging features only in their prognostic models. These included an SVM model in a multi-institution radiomics study (*n* = 80) which achieved PFS prediction with AUC values between 0.82–0.88 [116], and an RF model from a smaller (*n* = 40) radiomics study based on TCGA data which reported similar AUC values for PFS (0.8537) and OS (0.8554) [129].

Nineteen studies integrated imaging features with other data types: clinical and molecular data (*n* = 10), clinical only (*n* = 8), molecular only (*n* = 1). Age was the most commonly utilized clinical data type and was specified as being a selected feature in 12 studies.

### 3.6. Validation and Data Sharing Strategy

One crucial step in closing the first translational gap is to deploy a suitable validation strategy that uses an independent dataset. Resampling approaches can be applied to validate models with limited samples of data. 

#### 3.6.1. Breast Cancer

Eleven studies have not deployed any validation strategies. For the remainder, the radiomics features were validated in an internal test set, the most commonly used technique was the k-fold cross-validation (15/34) followed by leave-one-out cross-validation approach (5.8%—2/34). Other approaches included Cox regression to select the features correlated with PFS (2/34), grid search method or calibration of receiver operating characteristics (ROC) curve using Hosmer–Lemeshow test. External validation was performed only in a small proportion of the studies (3/34) with one study examining inter- and intra-observer assessments variability [48]. The code used for analysis was made available only for one study [22]. 

#### 3.6.2. Rectal Cancer

Out of the 15 papers, 13 included a clear internal validation method (these included 10-, 5-, 4 and 3-fold cross validation, leave-one-out cross validation, and bootstrapping methods). Of these, only four also tested their models on an external dataset. AUC was the most commonly quoted metric of performance (14 papers). In total, 10 of the studies used models including clinical data, and only one of the 15 studies included molecular data, in the form of circulating tumor cells (CTCs) and EVs such as cancer microparticles (MPs) [63]. None of the studies reviewed contained models or tools which had received regulatory approval for clinical use.

None of the 15 studies included made the code related to their models available. 

#### 3.6.3. Lung Cancer 

Overall, only 20% (*n* = 9) of papers performed external validation, with the remaining papers either being internally validated (*n* = 18) or not undergoing any testing of validity (*n* = 17). Internal validation approaches included validation using a portion of the test dataset with K-fold cross validation, leave-one-out cross validation and bootstrapping, or validation in an independent internal sample. One study [77] separated its sample population by image matrix size, such that one matrix size was used in the test dataset, and one was used in the cohort, to test validity across matrix sizes. The rarity of external validation, combined with the rarity of prospective studies (*n* = 5) or studies incorporating multi-center data (*n* = 11) is reflected by the fact that none of the models assessed have been approved by regulatory bodies for use in a clinical context.

#### 3.6.4. Glioblastoma Multiforme

Twelve manuscripts included single-institution datasets with fewer cases (range 15–181, mean 94.5), while four studies utilized larger datasets from two to six separate institutions (range 80–837, mean 288). Seven manuscripts focused entirely on publicly available data (TCGA/TCIA/BraTS17/BraTS18/Ivy-GAP), in addition to five and three papers which combined single and multi-institution data with open-source datasets respectively. 

Overall, 26 out of 31 studies outlined a clear internal validation method (typically with 10-, 5- or 3-fold cross-validation, leave-one-out cross-validation or bootstrapping), and a further 10 studies were able to externally validate their findings in an independent cohort. Hazard ratios were the most commonly reported model performance metric (16 papers), followed by area under ROC curve (AUC, *n* = 12), Harrell’s C/concordance index (*n* = 10), accuracy (*n* = 7), sensitivity/specificity/positive or negative predictive value (*n* = 7) and integrated Brier score/mean prediction error rate (*n* = 4).

## 4. Discussion

FDA/EMEA guidance documentation details the rigorous steps required in the route to the regulatory biomarker qualification of Artificial Intelligence and Machine Learning (AI/ML)-endpoints (FDA-qualified medical device development tool (MDDT)) [141,142]. Our review offers a snapshot of the existing literature in studies of radiomic/volumetric-based predictive or prognostic models, from which three areas stand out as key priorities in bringing volumetric-based and radiomics-based clinical trial endpoints closer to prime-time clinical implementation, namely: segmentation, validation and data sharing strategies. We have evaluated tumor types with the highest number of publications (i.e., breast cancer, rectal cancer, lung cancer and glioblastoma) to illustrate the common challenges in implementing volumetric and/or radiomic endpoints in clinical trial design. Although different biological processes characterize these diverse tumor types, the challenges for implementing novel imaging biomarkers are likely to be similar.

### 4.1. Strategy of Segmentation

Classical radiomics requires segmentation of a region of interest from which radiomic features are computed [143]. Manual segmentation by expert readers is time consuming and has the potential to introduce reader-dependent bias [11,110,143]. It is possible to assess the inter-reader variability of the resulting features and exclude those that are variable [48,143]. However, this may affect the predictive ability of the resulting tool, as tumor margin and peritumoral areas have been found to be salient features for predicting tumor response [79]. 

Large sample sizes are required especially for training deep learning models; however, the associated increase in workload with manual segmentation creates a key bottleneck. The ideal strategy for segmentation in the context of large datasets would be reliable, reproducible and high throughput. Automated or semi-automated tools for segmentation are being developed and utilized by researchers to relieve this bottleneck [144]. The increased sample size accessed through these approaches allows researchers to test and develop more data hungry predictive tools such as convolutional neural networks [110,145].

Using automated tools to segment large quantities of data is a tempting solution to the bottleneck; however, AI segmentation is vulnerable to artefacts and noise [146]. It is important to keep this in mind when developing data sharing infrastructure. Archives should be curated to identify images that may be affected by noise or artefacts so that teams developing the tools can apply appropriate pre-processing steps or exclude data to eliminate the risk of bias [147]. On the other hand, the increased availability of publicly available datasets is aiding the development of these tools [144].

It is now possible to create a fully automated radiomics pipeline, resulting in a multiparametric signature with superior performance compared to fixed-parameter radiomics signatures and conventional prognostic factors [120]. The model created by Li et al. [120] utilized automated harmonization techniques such as resampling, automated segmentation and radiomic analysis. Unfortunately, it was limited by its small sample size, and retrospective nature [120], further highlighting the need for accessible data warehouses.

Furthermore, segmentation of tumor sub-volumes may also be of importance to model performance [109,112,129], as exemplified by manual segmentation of GBM subregions, with each subregion yielding a different AUC (tumor only: 52.99%, edema only: 61.77%, necrosis only: 63.85%, all three subregions: 66.99%) [129].

### 4.2. Validation Strategy

Validation of a predictive model is a useful indicator of its potential effectiveness in the target population, and this is reflected by its inclusion in the RQS [11]. The TRIPOD statement outlines the different types of validation methods available to researchers developing prognostic tools [148]. Both TRIPOD and the RQS rank external validation as the most effective form of validation [11,148], as the data is more independent [11]. Unfortunately, external validation is rare in the literature while internal validation is more common [149]. This lack of validation of predictive models is a barrier to their implementation in clinical practice [143]. 

External validation can be achieved using data from other institutions directly [68]. This allows initial validation but can result in a small validation dataset requiring further study to characterize [68]. Alternatively, some groups opt to create test and/or validation datasets using data from an image biobank such as the cancer imaging archive (TCIA) which contains data from multiple centers [99].

External validation does not necessarily need to occur with the same imaging methodology, for example, some authors validated a previously published CT radiomic signature using a cone beam CT dataset [66]. Although this required modification of the signature to remove radiomic features that varied between CT and CBCT, it illustrates how external validation can be used to broaden the applicability of a predictive tool. 

An additional strategy to achieve external validation is to produce open-source code, along with a transparent methodology to allow independent verification of results by other research groups [11]. Requesting independent validation of an algorithm in an external dataset would be one option to externally validate a tool without the challenges of maintaining or using an image biobank. 

Finally, in the absence of an external dataset, some studies have utilized temporal validation, where the validation dataset is from the same institution but at a different timepoint [92]. Some studies have used temporal validation as a prospective test of AI tools [131]. Other studies have changed the image acquisition protocols to create a heterogenous dataset to validate the tool against [112]. Similarly, Park et al. separated its sample population by image matrix size, such that one matrix size was used in the test dataset, and one was used in the cohort, to test validity across matrix sizes [77].

### 4.3. Data Sharing

Open data models are necessary to improve discovery and reproducibility. For example, the RECIST measurements were validated by the European Organization for Research and Treatment of Cancer (EORTC) data warehouse of case report forms (CRF) from 50 clinical trials, with more than 6500 patients treated with chemotherapy and more than 23,000 patients treated with targeted agents [150]. However, creating a data warehouse of case report forms containing longitudinal measurements instead of primary imaging data makes it difficult to assess inter-and intra-observer variability and allow for external validation studies [150].

Sharing of imaging and clinical data can contribute to the development of AI tools by enabling them to use larger sample sizes, externally validate their results and incorporate more variables in their predictive tools [11]. However, there are some challenges to implementing and using such bio-banks including confidentiality, data ownership and curating the data [145]. Radiomic signatures are sensitive to changes in technical elements such as voxel size, use of contrast and other technical variations [120], so it is important that this data is also included in biobanks to allow data harmonization or exclusion [99].

Other strategies to acquire data include using clinical trial data as the development [53,112], or validation [95] datasets. Utilizing clinical trial data has the added benefit of enabling prospective testing of the tool. Papers that utilize clinical trial data either gather data from trials conducted within their institution [82] or acquire data from data warehouses such as TCIA [95]. Unfortunately, prospective studies utilizing clinical trial data not sourced from data warehouses rarely mention any data sharing methodologies, with some papers offering access to the original data through writing to an author [112]. The reluctance to make data openly and publicly available could be due to the factors such as consent and data security, for example Lou et al. specifically request that the data are used only for the purposes outlined to the data provider, and mandate that redistribution of the data is prohibited [79].

TCIA is a data warehouse that uses encryption along with a semi-automated de-identification procedure which includes manual review for each image prior to publication, to check for missed patient details in the file, such as pixel embedded patient information [151]. Semi-automated and visual quality control procedures are also in place to ensure images are visible and un-corrupted [151]. The requirement for manual checks means that TCIA must employ staff to grow and maintain the archive, as well as assist researchers in uploading data to, or using data from the archive [151]. As TCIA aims to be a diverse archive of varying modalities, acquisition protocols and level of prior de-identification mean that curation of data uploaded TCIA is a labor-intensive process. This limits the amount of data that can be processed, uploaded and made publicly available at one time.

Further challenges to data sharing and curation arise from radiomic approaches such as deltaomic analyses which require images from multiple timepoints. Furthermore, multiomic tools require not just imaging data but associated clinical or genetic data. Despite these challenges, projects such as the cancer genome atlas (TCGA) endeavor to create publicly available ‘next generation’ data warehouses with clinical, radiological, genomic and other data types for various tumor types [145]. 

Other relevant approaches, such as the large public–private partnership Vol-PACT (Advanced metrics and modelling with Volumetric Computed Tomography for Precision Analysis of Clinical Trial results) [152]. The Vol-PACT effort aims to develop volumetric-based CT metrics for precision analysis of clinical trial results in measurable solid tumors (non–small-cell lung cancer, colorectal cancer, renal cell cancer, and melanoma) [152]. Its initial benchmarking of the volumetric and mathematical modelling is by RECIST v1.1, immune-related RECIST, and case report forms (CRFs) with a goal to establish a repository of semi-automated calculated tumor burden [152]. To date, the results of Vol-PACT approach has been published for melanoma trials [153] and metastatic colorectal cancer trials [154]. The random forest algorithm of imaging features from 575 advanced melanoma patients outperformed the standard RECIST v1.1 response method [153]. The DL scores of imaging data collected during metastatic colorectal cancer VELOUR trial (NCT00561470) showed a better performance compared to sized-based criteria (RECIST v1.1 and early tumor shrinkage (ETS)) [154].

Various organizations have attempted to provide guidance for researchers using patient data to develop AI tools. Notably, the Joint European and North American Multisociety Statement on the ethics of AI highlight the need for these tools to minimize harm, ensure any potential harm is equally distributed amongst stakeholders and to curtail bias [155]. Despite this, it is known that there is some asymmetry in the racial representation of imaging archives, for example, TCGA overrepresents Caucasian populations despite the fact that there are some tumor types with different behavior in other ethnicities [147]. The multisociety statement acknowledges the need to develop policy for ethical practice for researchers developing AI [155]. Such policies have since been created, for example, the Analytic Imaging Diagnostics Arena (AIDA) in Sweden has created a grassroots policy informed by various stakeholders to guide AI researchers, taking into account legal frameworks such as GDPR [156]. This has led to the creation of the AIDA data hub, a national level data warehouse, which can be accessed by researchers upon request [156]. Both the AIDA and the multisociety statement acknowledge that new ethical challenges mean that best practice for data sharing is evolving, so it is important to reassess regulations, policy and advice to keep up with this change [155,156]. 

Increased availability of data may also allow research groups to use the increased sample size to utilize approaches that demand more data, such as convolutional neural networks [145], or increasing the number of radiomic markers included in a model [11], as, without an appropriate sample size, these approaches may be vulnerable to issues such as overfitting [143]. In parallel with improving the AI approaches, it is essential to create the framework for wider and free data sharing as recommended by the CRUK data sharing guidelines [157]. 

## 5. Conclusions

Current validated imaging biomarkers based on RECIST v1.1 were developed using a large collection of CRFs. The extraction of quantitative features from standard of care imaging for prognostic or predictive modelling is an active area of research that is rapidly generating potential imaging-based biomarkers of treatment response and outcome in oncological research. To date however, radiomics or volumetric biomarkers have been assessed in relatively small study cohorts and have shown varying degrees of promise. Ultimately, these approaches need to be tested within prospective, large multicenter studies before being ready for wide-scale adoption in clinical practice. Future use-cases will likely require (a) higher throughput systems for automated segmentation, (b) a higher standard for publishing externally validated models, (c) improved imaging data sharing capabilities to allow more prospective studies with volumetric/radiomics endpoints and (d) approaches for integration of radiology-based data with multiomic data. A robust and effective ethical and legal framework for data sharing will be crucial to support the discovery and validation of novel cancer imaging biomarkers towards the aim of regulatory biomarker qualification and clinical adoption.

## Figures and Tables

**Figure 1 cancers-14-05076-f001:**
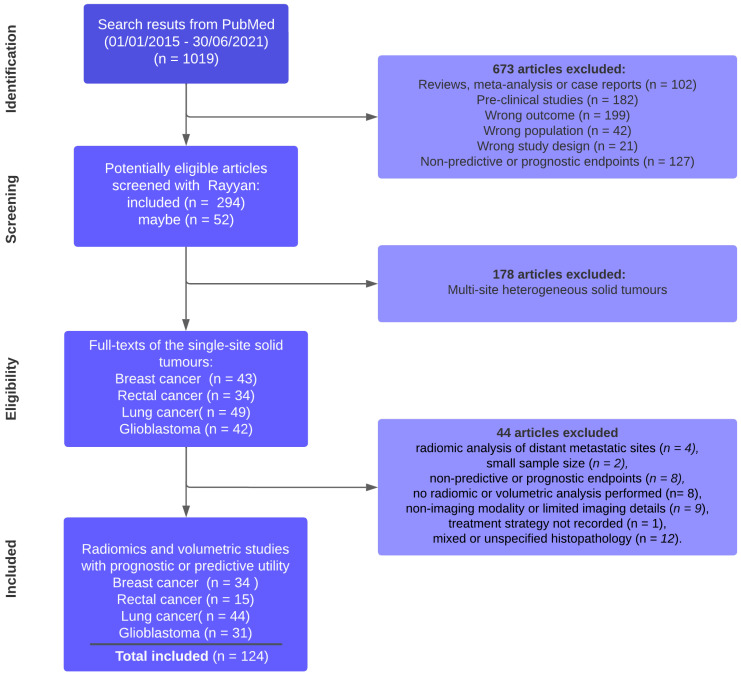
Flow diagram of studies selection process.

**Figure 2 cancers-14-05076-f002:**
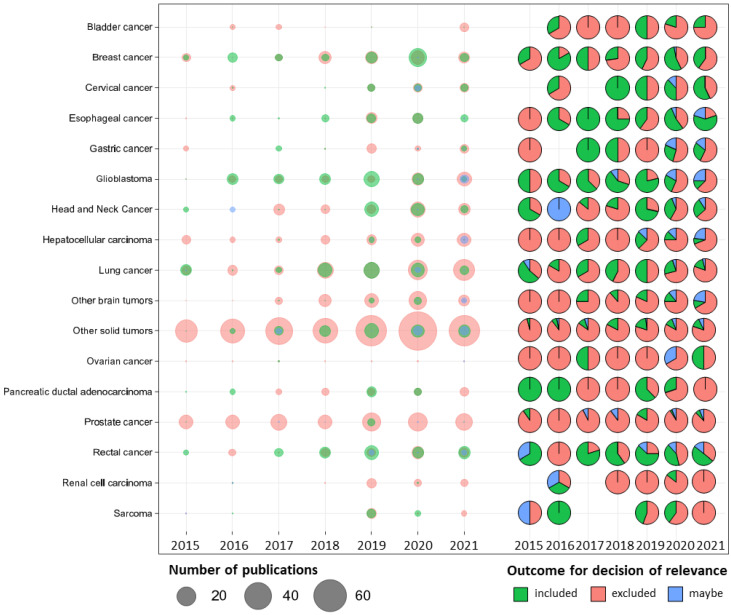
The outcome of the decision by year of publication. The bubble chart summarizes the relationships between tumor types, year of publication, number of publications and outcome of the relevance of assessment at abstract screening. The size of the dots represents the number of publications and the colors of the outcome of the decision after assessing the relevance of the publications’ abstracts. The scatter pie chart summarizes the proportion of papers in each relevance category.

**Table 1 cancers-14-05076-t001:** Summary of imaging modality and study design (* = USS; ** = MRI and CT, *n* = 1; MRI and PET-CT *n* = 2; PET-MRI and CT *n* = 1; *** = Cone beam CT; **** = PET-MRI (*n* = 1) and FET-PET (*n* = 1).

Tumor Type	Total Number of Papers	Proportion with Elements of Prospective Study Design	Proportion with Multicenter Data	Imaging Modality			
				MRI	PET/CT	CT	Other
Breast cancer	34	8	8	25	4	2	3 *
Rectal cancer	15	3	6	10	0	1	4 **
Lung cancer	44	5	11	0	15	29	1 ***
Glioblastoma	31	6	7	29	0	0	2 ****

**Table 2 cancers-14-05076-t002:** Summary of primary endpoints. Classification of surrogate endpoints is based on definitions from FDA’s surrogate endpoint table [2].

	Non-FDA Surrogate Endpoints	Pathology-Based Surrogate Endpoints (+/−RECIST Endpoints)	RECIST-Based Surrogate Endpoints	Survival Endpoints (+/− RECIST Endpoints)	Total
**Breast cancer**	**1**	**28**	**4**	**1**	**34**
Disease-free survival [21,27,32,47]	0	0	4	0	4
Overall survival [44]	0	0	0	1	1
Pathologic complete response [15,16,17,18,19,20,22,26,28,29,31,33,35,36,37,38,39,40,41,45,46,48]	0	22	0	0	22
Pathologic complete response + Disease-free survival [24,25,30,34,42,43]	0	5	0	0	5
Pathologic complete response + Durable objective overall response rate [34]	0	1	0	0	1
Predictive therapy response [23]	1	0	0	0	1
**Rectal cancer**	**14**	**0**	**1**	**0**	**15**
Disease-free survival [58]	0	0	1	0	1
Pathologic complete response [49,50,51,52,53,54,55,56,57,59,60,61,62,63]	14	0	0	0	14
**Lung cancer**	**1**	**0**	**17**	**26**	**44**
Disease-free survival [83,84,89,102,105]	0	0	5	0	5
Durable objective overall response rate [90,104]	0	0	2	0	2
Overall survival [64,66,69,71,72,76,78,82,85,91,93,94,97,98,100,101,107]	0	0	0	17	17
Overall survival + Progression-free Survival [70,73,75,80,86,87]	0	0	0	6	6
Overall survival + Disease-free survival [88,92,96]	0	0	0	3	3
Pathologic complete response [103]	1	0	0	0	1
Progression-free survival [65,67,68,77,79,81,99,106]	0	0	8	0	8
Progression-free survival + Longterm benefit [74]	0	0	1	0	1
Progression-free survival + Radiation pneumonitis [95]	0	0	1	0	1
**Glioblastoma**	**2**	**0**	**2**	**26**	**31**
Associations with biological processes [136]	1	0	0	0	1
Overall survival [109,110,111,113,114,117,118,119,120,121,122,123,125,127,130,132,133,134]	0	0	0	18	18
Overall survival + Progression-free Survival [108,115,126,128,129,136,137]	0	0	0	8	8
Progression-free survival [112]	0	0	1	0	1
Progression-free survival + pMGMT status [116]	0	0	1	0	1
Recurrence site [124,138]	2	0	0	0	2
**Grand Total**	**19**	**28**	**24**	**53**	**124**

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
