# Peer review of "Radiomic and Volumetric Measurements as Clinical Trial Endpoints—A Comprehensive Review"

_cancers, 2022, doi:10.3390/cancers14205076_

Round 1
Reviewer 1 Report
This review aims to identify the challenges and opportunities for discovering new imaging-based biomarkers based on radiomic and volumetric assessment. With the development of new therapies, with different ways of action, incorporation of integrative response and outcome biomarkers may help to predict treatment response.
The review is very complete. I only have minor comments.
I am not sure the title is appropriate. As explain by authors in the discussion part, this is more a snapshot of the existing literature in studies of radiomic/volumetric-based predictive or prognostic models, with a focus on 4 localisations. This should be more clear in the title.
Abstract:
Could you add the four tumor types studied in the paper, in the abstract and in the simple summary?
Results:
Do you think you can sum-up results regarding study design, type of imaging modality, and type of algorithms & primary endpoints (…) in one or two table to help the reader?
Author Response
Dear Sir/Madam,
I would like to thank you for the opportunity to submit our revised manuscript entitled “Radiomic and volumetric measurements as clinical trial end-points - ready for prime-time?” by Ionut-Gabriel Funingana, Pubudu Piyatissa, Marika Reinius, Cathal McCague, Bristi Basu and Evis Sala.
We have answered the reviewers’ comments, and you can find below a point-by-point list.
Title - Current titile: Radiomic and volumetric measurements as clinical trial end-points - ready for prime-time?
Proposed title: Radiomic and volumetric measurements as clinical trial end-points – a comprehensive review.
Abstract: Could you add the four tumor types studied in the paper, in the abstract and in the simple summary?
Thank you for your valuable suggestion. We have amended the abstract and the simple summary.
Results: Do you think you can sum-up results regarding study design, type of imaging modality, and type of algorithms & primary endpoints (…) in one or two table to help the reader?
Thank you for your valuable suggestion. We have tabulated the data and included it into two tables.
Yours sincerely
Dr Ionut-Gabriel Funingana MD - Clinical Research Associate in Ovarian Precision Cancer Medicine
Reviewer 2 Report
The authors performed a comprehensive literature search in application of radiomics as potential quantitative metrics for clinical trial design. Overall, the manuscript is well written and provides a concise but comprehensive review of the literature. However, there are major limitations:
- Focused evaluation of tumor types with highest number of publications (i.e. breast cancer, rectal cancer, lung cancer and glioblastoma) can be misleading since these are inherently and biologically different disease processes.
- There is limited information on difference in feature selection methodologies and statistical creation of radiomics signature and machine learning models involved in them
- There is limited reference to any article on direct comparison of RECIST and radiomics features
Author Response
Dear Sir/Madam,
I would like to thank you for the opportunity to submit our revised manuscript entitled “Radiomic and volumetric measurements as clinical trial end-points - ready for prime-time?” by Ionut-Gabriel Funingana, Pubudu Piyatissa, Marika Reinius, Cathal McCague, Bristi Basu and Evis Sala.
We have answered the reviewers’ comments, and you can find below a point-by-point list.
Suggestion 1 - Focused evaluation of tumor types with highest number of publications (i.e. breast cancer, rectal cancer, lung cancer and glioblastoma) can be misleading since these are inherently and biologically different disease processes.
Answer:
Thank you for your valuable suggestion. We have included these diverse tumor types with the highest number of publications to illustrate the common challenges in implementing volumetric and/or radiomic endpoints in clinical trial design. Although different biological processes characterise these tumor types, the challenges for implementing novel imaging biomarkers are likely to be similar. We have included this in our discussion
Suggestion 2 - There is limited information on difference in feature selection methodologies and statistical creation of radiomics signature and machine learning models involved in them
Answer:
Thank you for your helpful suggestion. The review aimed to provide an extensive explanation of the three key areas that require further improvements before integrating volumetric/radiomic-based imaging biomarkers. Therefore, we have decided to limit the information about other areas, which would have extended our review beyond the initial objectives.
Suggestion 3 - There is limited reference to any article on direct comparison of RECIST and radiomics features
Answer:
Thank you for your suggestion. We have cited the papers that used RECIST v1.1 based endpoints alone or in combination with other validated clinical trial endpoints. Therefore, we have summarised all classes of surrogate endpoints and cited the relevant papers in Table 2.
Thank you again for your consideration of our revised manuscript.
Yours sincerely
Dr Ionut-Gabriel Funingana MD
Honorary Clinical Research Fellow in Medical Oncology